# Information from Noise: Measuring Dyslexia Risk Using Rasch-like Matrix Factorization with a Procedure for Equating Instruments

**DOI:** 10.3390/e25121580

**Published:** 2023-11-24

**Authors:** Mark H. Moulton, Brock L. Eide

**Affiliations:** 1Pythias Consulting, Vancouver, WA 98664, USA; markhmoulton@gmail.com; 2Neurolearning SPC, Edmonds, WA 98026, USA

**Keywords:** dyslexia, matrix factorization, Rasch model, alternating least squares, test

## Abstract

This study examines the psychometric properties of a screening protocol for dyslexia and demonstrates a special form of matrix factorization called Nous based on the Alternating Least Squares algorithm. Dyslexia presents an intrinsically multidimensional complex of cognitive loads. By building and enforcing a common 6-dimensional space, Nous extracts a multidimensional signal for each person and item from test data that increases the Shannon entropy of the dataset while at the same time being constrained to meet the special objectivity requirements of the Rasch model. The resulting Dyslexia Risk Scale (DRS) yields linear equal-interval measures that are comparable regardless of the subset of items taken by the examinee. Each measure and cell estimate is accompanied by an efficiently calculated standard error. By incorporating examinee age into the calibration process, the DRS can be generalized to all age groups to allow the tracking of individual dyslexia risk over time. The methodology was implemented using a 2019 calibration sample of 828 persons aged 7 to 82 with varying degrees of dyslexia risk. The analysis yielded high reliability (0.95) and excellent receiver operating characteristics (AUC = 0.96). The analysis is accompanied by a discussion of the information-theoretic properties of matrix factorization.

## 1. Introduction

The first clear report of developmental dyslexia appeared in the medical literature in 1896, when British ophthalmologist W. Pringle Morgan described a fourteen-year-old boy named Percy, who despite receiving seven years of “the greatest efforts to teach him to read”, could still only read and spell at the most basic level, even though his schoolmasters believed him “the smartest lad in the school” [1]. Subsequent efforts to precisely define dyslexia and clarify its core causes and features have followed a tortuous path. Attention initially focused on possible visual deficits, later shifted toward deficits in the phonological processing system, and over the last several decades, broadened to recognize that dyslexia is multifactorial in origin [2,3,4,5]. Multiple independent risk factors have been implicated in the etiology of dyslexic reading and spelling challenges at the cognitive processing level, including phonological processing, visual attention, working memory, naming speed, processing speed, and implicit learning, as have environmental factors. Genetic studies have revealed a similar heterogeneity, with a recent study revealing 42 genetic loci significantly associated with dyslexia [6]. Not surprisingly, great variability in symptoms can be observed at the clinical level, in which individuals with different blends of risk factors may display quite different clinical presentations, responses to interventions, and long-term prognoses.

This inability to identify a single source or precise clinical presentation for dyslexia has led some to propose that the concept of dyslexia itself is both vacuous and unnecessary [7], or that it simply denotes poor reading achievement by another name. However, these contentions ignore a substantial body of research literature on dyslexia that has repeatedly identified a core group of subjects that, in the words of the World Federation of Neurology [8], share the common and readily recognizable experience that “despite conventional classroom experience, [they] fail to attain the language skills of reading, writing and spelling commensurate with their intellectual abilities”. This discrepancy between general and, especially, verbal ability and reading achievement, particularly in the domains of decoding, spelling, and reading fluency, remains a key component of the dyslexia concept [9]. Additional core features of the emerging consensus on the dyslexia diagnosis include the stipulations that the core functional challenges in dyslexia consist of word- and sub-word-level difficulties with decoding, spelling, and reading fluency and that these challenges are produced by the kinds of independent risk factors that produce the lower-level processing challenges cited above.

At a more general level, the case can also be made that skepticism about the dyslexia diagnosis often displays confusion about the nature of clinical syndromes. Syndromes are combinations of symptoms that can be produced either by a single discrete cause or, as is the case with dyslexia, occur so commonly together that they constitute a distinct clinical picture [10]. In clinical medicine syndrome, diagnoses of this latter type are of immense practical value and are often used in patient care, as for example with such syndromes as acute shock, respiratory distress syndrome, stroke, hypertension, and end-stage renal disease. What such syndromes have in common is that each results from a varied group of causes yet results in a distinct and recognizable set of clinical consequences. The treatment of such syndromes typically requires both the management of these common downstream consequences, as well as the identification and management of the discrete upstream sources to prevent further damage or promote improvement.

Attempts to measure, model, define, and diagnose clinical syndromes will inevitably fail if relevant information is neglected or if the measurement techniques employed fail to adequately sample and represent the data. In the case of dyslexia, the intrinsic complexity and multidimensionality of dyslexia has thus far hindered the identification of a single measure or clinical result that can be used to make a diagnosis [4]. Many attempts to devise simple screening tools have failed to sufficiently represent diagnostic and theoretic core features, such as ability–achievement discrepancy, multifactorial causality and presentation, and the need for signs of abnormal functioning in both low-level cognitive functions and reading achievement. In addition, in current clinical practice, there is no single universally agreed upon gold standard assessment test or measure used to diagnose dyslexia. Instead, dyslexia is typically diagnosed by experts with training and experience in relevant fields of cognition or learning using a variety of testing instruments that contain multiple dimensions and a wide range of item types. Each clinician has his or her own favored test instruments and cut scores, and a positive diagnosis of dyslexia by one clinician may not generalize to other clinicians.

In light of this background, this paper has four objectives: (1) to present a screening instrument of sufficient breadth and depth to capture the complexity of dyslexia; (2) to present a method of analysis, matrix factorization, that is suitable for analyzing such data; (3) to demonstrate how matrix factorization can be used to construct a scale, the Dyslexia Risk Scale (DRS), that meets the requirements of objective measurement as defined in psychometrics in the Rasch model, such that examinees can be compared even though their measures are based on responses to different but overlapping sets of items; (4) to argue for universal age-independent diagnostic standards implemented through a community bank of equated matrix-factored “item coordinates”. It is also intended to present several information-theoretic properties of matrix factorization.

### The Problem of Multidimensionality

When screening for dyslexia, dyslexia risk is usually quantified as a single raw score summed across a person’s responses to items on a multidimensional instrument. The result is a composite score—composite across dyslexia dimensions—that is prone to change meaning when items are removed or added to the instrument over time, complicated by the fact that items have not only different sensitivities to the various dyslexia dimensions but different difficulties (see [11] for an example of a composite dimension dyslexia screener). Persons who take the instrument can usually only be compared to others in the same age group who take the same instrument.

This constraint seriously restricts the ability to compare persons across instruments, follow their progress over time, or compare persons from different age groups because instruments can be effective only if they are age appropriate. Without a technique to equate multidimensional instruments across age groups, no age-sensitive comparison is valid and the longitudinal tracking of dyslexia over time becomes infeasible. Given these and other psychometric difficulties, composite raw scores drawn from multidimensional instruments are not suitable for developing a valid and generalizable community gold standard.

Educational psychometrics has developed its own way of addressing similar issues under the rubric of item response theory (IRT) [12]. Instead of summing scores across constructs, items are grouped by construct, e.g., Math and Reading, and analyzed separately with models that require some degree of unidimensionality across items. The Rasch model is the simplest and strictest of the IRT models in its requirement for unidimensionality as a condition of fit [13,14]. Unidimensionality makes it possible to “equate” instruments for a specific construct to make it possible for students to be compared across reading instruments, for example, as if they had all taken the same instrument [15,16]. A widely used equating technique is to use common items to adjust for the relative difficulty and variance of different instruments and place all students on a common “scale score” metric that transcends the raw scores of individual instruments [17]. If desired, a composite score can be calculated as a weighted sum of construct scores that, like its component scores, will be psychometrically stable.

While calculating composite scores from multiple unidimensional instruments has proven practical for large-scale assessments, it is far from a definitive or general solution. It requires human expertise to assign items to constructs and only works when the items within each construct are sufficiently unidimensional—sensitive to person variation in one and only one dimension. This is never precisely the case because every item inevitably requires multiple skills as a condition of success; a word problem requires skills in both math and reading. However, in the educational assessment domain, items assigned to a construct tend to be sufficiently similar—and the students assessed have been exposed to sufficiently similar curricula—that the requirement of unidimensionality is approximated to a degree that supports test equating and generalized educational measurement.

Note that multidimensionality has two senses. There is multidimensionality *between* items or item groups (math items versus reading items) and multidimensionality *within* each item (math and reading as two skills that are both required for success on each of a set of word problems). This distinction has mathematical importance. “Between-item” multidimensionality such as that encountered in educational assessments can be handled using a series of unidimensional analyses or, more powerfully, with a between-item multidimensional model able to trade information between subscales. “Within-item multidimensionality”, such as that encountered when trying to predict, for example, movies a person might like, requires multidimensional models, one of which is the matrix factorization model described in this paper. Unidimensional models, no matter how they are applied, will not work with such data.

Matrix factorization has proven its effectiveness in the field of machine learning, most famously in the 2009 Netflix Prize for predicting movie ratings, where, despite its simplicity, it regularly occupied a high position on the Leader Board and was a major contributor to the winning entry [18]. Because dyslexia items are often sensitive to multiple dyslexia risk factors at the same time, and because individuals often differ markedly in the types of challenges they experience, dyslexia datasets are, in theory, more like movie preference datasets than educational assessment datasets, suggesting that matrix factorization may be an appropriate way to analyze them [19]. For example, multiple cognitive factors, as mentioned above, have been shown to independently increase the risk of dyslexic reading and spelling difficulties, and many dyslexia authorities have suggested that the identification of dyslexia risk should be based upon multifactorial risk identification models [2,3,4,5]. The development of the screening tool described in this paper was based on this theoretical premise.

It should be noted that matrix factorization is by no means new. It is, in essence, another version of principal components analysis (PCA), invented in 1909 by statistician Karl Pearson and still used widely as a workhorse of machine learning. However, PCA is usually applied to square covariance matrices, not rectangular data matrices, with a focus not on prediction or measurement but on finding the latent factor structure. Applied to rectangular data matrices, it becomes singular value decomposition (SVD). However, SVD and PCA assume complete data; the algorithms used in matrix factorization do not. This turns out to be an essential difference, making it possible not only to analyze sparse datasets but to actualize objectivity and apply an abstract pool of person and item coordinates to datasets well beyond the initial calibrating dataset.

From an information-theoretic perspective, it has been proven that when certain assumptions are met, particularly independent Gaussian noise, PCA and, therefore, matrix factorization minimizes information loss when dimensionality is reduced [20]. This will be discussed further in Section 4.3.

In applying matrix factorization to a dyslexia dataset, there is still the question of how to equate instruments to compare persons across instruments, age groups, and over time. Fortunately, the equating techniques developed for educational assessment using the Rasch model transfer cleanly to matrix factorization (although they do not appear in the machine learning literature) and can be applied effectively to multidimensional datasets. The open-source matrix factorization software used in this paper, *damon.psymethods.nous_legacy*, referred to as “Damon” and written in Python 3.10, was originally developed for an educational assessment company, specifically to apply Rasch-like equating designs to multidimensional datasets, such as those associated with movies and other within-item multidimensional domains for which unidimensional models may not be appropriate [21].

This paper discusses a procedure for modeling multidimensional dyslexia data using Damon 0.11 to calibrate items and score persons in a way that conforms to the psychometric objectives of the Rasch model [22]. Because it differs in important respects from other matrix factorization procedures, it is qualified under the name “Nous” (Nous, the procedure implemented in Damon). Nous generates approximate equal-interval dyslexia risk measures that have a single conceptual meaning across the breadth of the scale—the “Dyslexia Risk Scale” (DRS). It also achieves construct invariance across item and person samples, permits the equating of dyslexia instruments using common items, makes it possible to compare persons of all ages who take age-appropriate instruments, and has the potential to measure dyslexia risk in terms of a common community scale.

## 2. Materials and Methods

### 2.1. Background

#### 2.1.1. Participants

From 2017 to 2019, a mobile computer application-based dyslexia screening instrument containing 577 items was administered to a diverse sample of 828 English-speaking persons ranging in age from 7 to 82 sampled to range in self-reported dyslexia risk from no suspicion of dyslexia (i.e., no self-reported perception of difficulties with decoding, spelling, oral reading fluency, etc.) to previous formal diagnosis of dyslexia (approximately half of the sample). 

Individuals took only those items appropriate to their age group (child, teen, adult), exposing them on average to roughly 76% of the item pool (443 items, including timings), though this varied by individual. There were 19 distinct item types intended to target different aspects of dyslexia. This included “timing” item types in which the length of time required to submit an answer to each item was recorded and used as an indicator of dyslexia risk.

#### 2.1.2. Instrument Subtests

There were nine subtests:

*Vocabulary*. A recorded voice narrator pronounces a target word. Users are asked to select the picture that best represents the word out of four alternatives.

*Auditory-discrimination.* A recorded narrator pronounces two words, then asks the subject to choose whether they are the same word or different.

*Visual matching.* The first group of items features a single row of six single letters, spaced to occupy the whole width of the screen. The next group features a single row of six letter pairs, with the paired letters occupying adjacent spaces and the pairs separated from each other at an equal distance to occupy the whole width of the screen. The last group of items feature a single row of six letter triplets with each letter of a triplet occupying adjacent spaces, the triplets spread out across the screen row. For each row, there are only two matching letters, letter pairs, or letter triplets. Examinees are asked to tap on the pairs.

*Rapid automatized naming (RAN).* In the first group of items, examinees are shown five rows of eight equally spaced tiles displaying one of five colors and are asked to name the colors as quickly as they can in a clear and understandable manner. The second group of items is similar to the first except that pictures are named.

*Phoneme elision*. The recorded voice narrator says the phrase, “What’s [*x*] without [*y*]?”, with x being a word and y being one of the sounds in the word. For the first group of items, the narrator presents two options for the resulting word from which the examinee chooses. For the second group, three options are presented.

*Working memory*. The narrator recites a list of items in sequence, after which the examinee is shown pictures of the items and asked to identify them in the order spoken. The list gradually becomes longer.

*Real word reading*. A real written word is briefly flashed on the screen. The narrator speaks three ways of pronouncing the word, from which the examinee chooses.

*Nonsense pseudoword reading*. The format is the same as the real-world reading subtest, but non-words (“nonsense” pseudowords) are used.

*Passage level reading*. A passage is displayed, and examinees are asked to read the passage aloud as quickly as possible while retaining accuracy. Upon completing each passage, it is hidden, and a group of multiple-choice reading comprehension questions is displayed.

Subtests were presented with gradations of difficulty and tailored to the age of the examinee.

The screening tool thus takes account of key factors in the emerging consensus on dyslexia, including identification of the ability/achievement discrepancy and differences in low-level processing functions that contribute independently to dyslexia risk, creating a model and set of measurements that sufficiently represent the relevant data regarding the nature, extent, and origin of the reading and spelling problems experienced by persons with the symptom complex called dyslexia. The data produced aim to represent the functional challenges of persons with dyslexia with a minimum of distortion or loss of information relative to their actual lived experiences.

#### 2.1.3. Data Characteristics

The dyslexia dataset contained a mixture of dichotomous correct/incorrect multiple-choice items and continuous clocked time intervals indicating time spent submitting a response. The “timing” items tended (as expected) to be negatively correlated with the dichotomous scores of the items with which they were associated. Data columns included respondent age, gender and other demographic characteristics, and self-reported dyslexia classification to ensure the sufficient representation of dyslexic and non-dyslexic persons in the sample. Because persons were only exposed to age-appropriate items, the data array was composed of responses to blocks of items with substantial overlap of common items across age groups and between adjoining age groups. Approximately 25% of the dataset consisted of blocks of missing data. The data array can, therefore, be interpreted as three age-appropriate datasets with common items merged into a single person-by-item data array with 25% missing data. The resulting dataset was used to “calibrate” the items in the instrument and build an automated scoring tool to be administered electronically on an as-needed basis.

#### 2.1.4. Clinician Ratings

In the absence of a diagnostic gold standard for dyslexia diagnosis, we employed an expert recommendation protocol. To specify a dyslexia risk construct, an expert in the clinical diagnosis of dyslexia reviewed all data collected for each person and assigned a rating on a scale from 0 (no dyslexia risk) to 10 (severe dyslexia risk) based on the data and a mix of quantitative and qualitative considerations. These included age and age group; raw score relative to peers on four “foundational” dyslexia constructs (phonological awareness, working memory, naming speed, and visual attention) and vocabulary and two reading achievement constructs; score “patterns” across subscales; and the length of time required to submit a response to each item. Clinician ratings were appended to the dataset as an additional column. Instead of contributing to the dyslexia measure as an extra “item”, the clinician ratings were used to specify the content of the dyslexia construct, to define what “dyslexia” *means*, and from that, to derive the Dyslexia Risk Scale (DRS). Methodologically, it plays a role in the matrix factorization procedure analogous to that of a dependent variable in a regression equation, in which latent factors are generated from the remainder of the dataset to serve as predictor variables, and the predictions themselves (suitably linearized) are the dyslexia measures. 

The procedure used for generating an expert rating of dyslexia risk was treated as a black box, not intended to be generalizable or a gold standard, just one expert’s best clinical guess based on available data. The variable is referred to as the “Clinician Rating”. Similar ratings were assigned to six dyslexia subscales and analyzed in much the same way. For simplicity, we focus on the primary dyslexia construct.

It is not necessary to define a dyslexia construct in terms of expert ratings. One could just as easily use some form of weighted sum of expected responses per person to derive a dyslexia risk measure. However, using a clinician has important advantages:It provides a reasonable expectation of construct validity, that measures based on these ratings will make sense and be interpretable in a clinical setting—so long as the ratings are successfully modeled using the remainder of the data.As ratings, clinician expert judgments are thus made *explicit*. In their absence, expert judgment is still required to decide how to weight and combine items to build a construct but in a way that is *implicit* as part of an instrument-specific process that is not as easy to describe or defend.An expert clinician has latitude to base ratings on experience and an informed feel for patterns of responses that may be missed by a conventional scoring method.By examining how well the instrument and model generate expected ratings that “fit” the clinician ratings, the degree to which instrument, data, and model are valid for dyslexia risk measurement can be quantified.It is straightforward to refine or rebuild the dyslexia construct by convening a panel of qualified clinicians, each of whom assigns his or her own ratings to each examinee in the dataset.

### 2.2. Matrix Factorization

To factor a rectangular matrix of observed values **X** with *N* rows and *I* columns means to calculate an *N* × *D* row matrix **R** and a *D* × *I* column matrix **C**, such that when the two are multiplied, they yield an *N* × *I* product “estimates” or “expected values” matrix **E** that maximally fits **X** given user-specified “dimensionality” *D*, where *D* < (*N*, *I*). *D* is the number of hypothesized “dimensions”, also called “factors”, equivalent to the “rank” of **R** and **C**. The difference **X**-**E** is the “residuals” or error matrix **ε**, also called “noise” and later, **Res**.
**X** = **RC** + **ε** = **E** + **ε**
(1)

Each cell in the estimates matrix **E** is the dot product (sum of products across dimensions) of the corresponding row and column entity vectors:**E***_ni_* = **R***_n_* **· C***_i_* = **R***_n_*_1_**C***_i_*_1_ + **R***_n2_***C***_i_*_2_ + … **R***_nD_***C***_iD_*(2)

The matrix decomposition algorithm used by Nous is a variant of alternating least squares (ALS). **C** is initialized with random numbers. Each row **R***_n_* person vector is calculated by applying ordinary least squares to the *non-missing* data in that row **X***_n_*_._[*non-missing*] using the corresponding columns in **C**. This yields the first iteration of **R**. Then, each column **C***_i_* is solved the same way, by applying least squares to the non-missing data **X**_._*_i_*[*non-missing*] in that column using the corresponding rows in **R**. For each iteration *t*, **R***_n_* and **C***_i_* are updated by least squares using only non-missing data and the corresponding non-missing portions of **R** and **C**, as follows:[**R***_n_*]*^t^*^+0.5^ = [(**C^T^C**)^−1^ **C^T^X***_n._*]*^t^*(3)
[**C***_i_*]*^t^*^+1^ = [(**R^T^R**)^−1^ **R^T^X***_.i_*]*^t^*^+0.5^(4)

Improving **R** and **C** iteratively by alternating least squares solutions, an acceptable degree of convergence is achieved, usually in less than 20 iterations. The columns in **R** (and rows in **C**) are linearly independent, though Nous takes an additional step with **R** to make them orthonormal, as well. Here, they are called “dimensional coordinates” because they describe a vector location in a space of *D* dimensions for each person and a vector location for each item. They are equivalent to what are called “embeddings” in the artificial intelligence world.

While this gives the essence of alternating least squares, the Nous implementation differs in several ways from other implementations. Instead of adding a regularization term to reduce overfit and avoid convergence problems and the degenerate solutions that can occur in rare cases, it adopts the simpler but more laborious procedure of forcing **R** to be orthonormal with each iteration using *QR* decomposition and performing the decomposition multiple times with different dimensionalities and random number seeds, picking the “best” seed and dimensionality for maximizing the “objectivity” statistic described below.

Nous also requires as a condition of fit to the model that all items occupy a “common space”, meaning that all have nonzero values on the same set of dimensions and zero values on all others. This removes the influence of all extraneous person dimensions by literally zeroing them out. This is directly analogous to the Rasch requirement that all items be tuned to a single dimension, and in this sense, Nous is stricter in its requirements than other matrix factorization methods that use, for instance, *ridge* or *lasso* regression to calculate regularization parameters. For both Rasch and Nous, if an item shows substantial misfit when comparing observations to estimates, it is deemed eligible for exclusion from the test, even if it is a good item in other respects. This requires more work for analysts and the employment of human judgment but results in more generalizable and *understandable* measures. Measures lose “objectivity” (discussed below) when the space in which they reside and from which they were calculated is not rigorously defined.

To obtain dyslexia measures, the **R** coordinates are treated as predictor variables of a normalized version of the clinician ratings. **R**’s dimensional variables are well-suited for multivariate regression due to their linear independence; each least squares calculation in ALS is, in fact, equivalent to a regression conditional on **R** or **C**. The resulting regression coefficients become “clinician coordinates” (labeled **C**[*clinician*] in Figure 1), a set of coordinates for each scale or subscale. The predictions for each cell, the expected values or cell estimates, become a preliminary form of dyslexia measure (labeled **DRS** in Figure 1). Therefore, for any new person *n* who takes some minimum number of items, his or her coordinates vector **R***_n_* is calculated and multiplied by clinician coordinates vector **C**[*clinician*] using Equation (2) to estimate a preliminary dyslexia measure **E**[*clinician*], which is rescaled to the more user-friendly DRS. We are, in effect, predicting how the clinician would rate that person *if the clinician were present*.

### 2.3. Equating and Reproducibility

A psychometric model is called “reproducible” if it can be shown that as a condition of fit to the model, it generates person measures that are the same regardless of the items the person takes and item parameters that are the same regardless of the persons who take them. Of the various item response theory models, the Rasch model and its derivatives are the only ones that have this property, also called “specific objectivity” [21]. Nous can be recast as a multidimensional generalization of the Rasch model, in a deterministic rather than probabilistic metric, and therefore, enjoys the same specific objectivity property so long as dimensionality *D* has been correctly specified. This can be seen by experimenting with simulated matrix **E** = **RC** at dimensionality *D*. If **E** perfectly fits the data **X** (noise is zero), when calculating the **R***_n_* coordinates for person *n* using least squares, any subset of items **C**[*sub*] used to model their corresponding data values **X**[*sub*] can be shown to yield exactly the same **R***_n_* solution. The same applies when calculating the **C***_i_* coordinates of item *i* using subsets of **R**. The only requirement is that the number of data values and coordinates used to calculate a least squares solution be greater than the number of dimensions used to model the data (an overdetermined solution); the more the better, depending on the noise in **X**. The objectivity property fades the more noise is in **X** (though the model tolerates a lot of noise) and to the degree dimensionality *D* is not correctly specified, particularly if *D* is too large.

It is this property that makes it possible for the Rasch model and Nous to be robust to missing data—effective even with extremely sparse datasets in which data are missing randomly or nonrandomly. There is no requirement, as with other statistical methodologies (including other factorization methods), to impute values for the missing cells as part of the estimation process. Both models sidestep missing cells entirely, working with whatever data are available. This property also means that they do not assume that persons and items are representative of a population, nor that they are normally distributed. One consequence is that the models are robust to small datasets (e.g., 100 × 100). All that is required is that the person sample be reasonably diverse, items span a common dimensional space, and the correct dimensionality be specified.

It is important to clarify the sense in which the **R** and **C** coordinates are reproducible. During estimation, they are initialized with random numbers; the choice of random numbers does not matter (much). That means the implied origin and axis orientations of the vectors in the final **R** and **C** matrices are entirely arbitrary and in that sense, are not at all reproducible. There is no attempt, as in factor or principal components analysis, to label the dimensions or rotate the coordinate system to maximize variation along one axis. 

To “equate” Form B to a reference Form A, and thus, generalize Form A’s coordinate system, one designates the **C** item coordinates of Form A as the “reference” coordinate system and uses it to “anchor” the **C** coordinates of the items on Form B that the two forms have in common, **C**[*common*]. When calculating **R** and **C** for Form B, instead of random numbers, **C**[*common*] items are initialized with the corresponding coordinate values of Form A. These are sufficient to calculate **R** for Form B and from these, the remaining Form B **C** coordinates, as well, in one least squares iteration. This places all persons and items for both instruments into a common coordinate system so that mutual comparability is achieved (Figure 2). 

If the **C**[*common*] item coordinates from Form A are grafted successfully onto Form B, such that the Form B cell estimates for those columns “fit” the corresponding observations, then we can say that **R** and **C** have been “equated” across the two instruments. This anchoring technique is virtually the same as that used in the Rasch model to equate unidimensional tests but generalized to higher dimensional spaces—all made possible by the special objectivity property and strict specification of a common space.

### 2.4. Dimensionality

#### 2.4.1. Optimal Dimensionality

In matrix decomposition, the most important unknown is dimensionality *D*. If *D* is set too low, **E** (and **R** and **C**) may be reproducible but miss important sources of variation and yield poor predictions of cell values. If *D* is set too high, the estimates matrix **E** will be similar to **X** but biased by its noise **ε**, called “overfit”, causing the results to be unreproducible across subsets of the data matrix and other linked forms. Nous, therefore, uses two criteria to identify optimal dimensionality: (a) the ability to predict cell values that have been made missing (called “accuracy”); (b) the similarity of the **R** (or **C**) coordinates when calculated from different subsets of **X** (called “stability”).

The “accuracy” curve plots for each dimensionality *d* from 1 to *D*[*max*] the Pearson correlation *r***_X,E_***_,d_* between “pseudo-missing” observations and estimates for the same cells (equivalent to holding observations out of a “training sample” to predict performance on a “validation sample”):*Accuracy_d_* ≡ *r***_X,E_**_,*d*_ = *correl*(**X**[*pseudo*-*missing*], **E**[*pseudo*-*missing*])_*d*_(5)

The “stability” curve plots for each dimensionality the Pearson correlation *r*_1,2*,d*_ of **R** calculated from two non-overlapping samples of items (alternating columns).
*Stability_d_* ≡ *r*_1,2,*d*_ = *correl*(**R**[*item group* 1], **R**[*item group* 2])_*d*_(6)

The root product of *accuracy* and *stability* Nous calls *objectivity*:*Objectivity_d_* ≡ (*Accuracy_d_* ∗ *Stability_d_*)^1/2^(7)

Objectivity hits its maximum at a (usually) well-defined peak representing the “true” dimensionality, making it possible to identify unambiguously the optimal dimensionality *D*. At *D*, **R** and **C** are as reproducible, and **E** as accurate, as possible given the data.

Objectivity is an important concept that deserves a more rigorous mathematical foundation than what is presented here. For instance, Equation (7) omits the consideration of item independence, an important component of objectivity. Ideally, the formula should be derived solely in terms of standard error, but that depends on a standard error statistic that is valid even when dimensionality is incorrectly specified. Equation (5) is viewed as a necessary but provisional stand-in until a better objectivity statistic can be derived.

#### 2.4.2. Common Dimensionality

Damon’s matrix factorization is a “within-item multidimensional” model. Its objectivity properties, as well as the ability to identify an optimal dimensionality, depend on the degree to which every item on the instrument is sensitive to person variation on a common set of (often unknown) within-item dimensions and *insensitive* to all other person dimensions. When an item is sensitive to extraneous dimensions, it interacts with persons in anomalous ways identifiable as “misfit” between the observed and expected values for those data columns. Therefore, calibrating a test instrument requires “analysis of fit” in which items (and sometimes persons) with high misfit are removed from consideration for purposes of calibrating a reference bank of item coordinates, so long as such removal does not compromise the reliability of the test. This exactly parallels “analysis of fit” in Rasch analysis.

This way of thinking about data analysis—removal of items and persons to meet the requirements of a model—is philosophically the reverse of most statistical thinking [13]. Instead of looking for a statistical model that maximally fits the observed data, we are deliberately collecting and selecting data (items and persons) in a way that maximizes fit to an objectivity model—in this case a multidimensional linear generalization of the Rasch model set at optimal dimensionality *D*—as a consequence of which the resulting measures and parameters can be claimed to that extent to be objective and reproducible across datasets, so long as those datasets also fit the model.

### 2.5. Data Metric

Unlike Rasch, which models dichotomous and polytomous data using category probabilities, ordinary least squares assumes linear interval data with no upper or lower limits, and alternating least squares requires all columns to be in the same metric. However, our dyslexia dataset contains a mix of dichotomous (correct/incorrect) data and reverse-correlated continuous timings on a scale with a hard lower limit of zero, plus a polytomous rating scale for clinician ratings. Nous handles variation in column metrics by first converting all columns to a “pseudo-logit” metric. Logits (log-odds units) are a linear transformation of probabilities and, like z-scores, define an unbounded linear scale for which alternating least squares is well-suited. Because the probabilities needed to compute “true logits” are not at first known, the transformation assigns a “plausible” logit value to each data value depending on whether the data in the column are dichotomous, polytomous, interval, ratio, continuous, or discrete. Analysis is performed in the pseudo-logit space, in which cell standard errors are also calculated, given which several methods can be employed to derive “true logits” and probabilities for each cell, and from these, the original data metric can be recovered if desired.

For this dataset, pseudo-logit model estimates in the clinician rating column were converted to “semi-true” logits by rescaling according to the average measurement error of examinees to recalculate probabilities and logits. The resulting logits were rescaled linearly to create the DRS with a minimum near 0 and maximum near 100. However, unlike a bounded 0–100 scale, each unit indicates an approximately equal increment of increased dyslexia risk. Accompanying each DRS measure is a standard error statistic unique to that person.

It is important to note that, even with conversion to pseudo-logits, categorical data (especially dichotomous) do not meet some of the requirements of ordinary least squares. For instance, least squares assumes that the expected error per cell is homoscedastic, the same from cell to cell, whereas the errors per cell for dichotomous and polytomous data are expected to be heteroscedastic. A smaller residual is expected when cell estimates are close to the standardized upper or lower category values than when they are in the middle, between those values. However, an important property of ordinary least squares not shared by other estimation methods, such as maximum likelihood, is that least squares solutions (**R** and **C**) are not biased by heteroscedasticity. What is affected are secondary statistics, like variance. However, as will be seen in the next section, when least squares is applied in an alternating way to a *matrix* rather than a single column, it becomes possible to decompose the heteroscedasticity of error by row and column and calculate unbiased error statistics for each.

### 2.6. Estimating Uncertainty

Psychometric inferences require standard error statistics when evaluating and comparing individual person measures, so every effort has been made to calculate them. Several classes of methods are available for estimating error in this context, such as bootstrapping, analysis of the estimates covariance matrix, and jackknife estimation, but they require computationally expensive repeated sampling and estimation. What is needed is a simple cell standard error formula that can be applied quickly and efficiently. Because of the essential role of standard error in calculating reliability and information entropy, as discussed in Section 4.1, we take the time to present the Nous standard error equations, which we have not found presented elsewhere.

Given ambiguities in the statistical meaning of “error”, it is important to clarify what “standard error” means in this context. Consider how model-compliant data are generated. We start by simulating “true” **R***_True_* and **C***_True_* coordinate arrays, whose product is a rectangular array **T** (for “true”), then add Gaussian noise to simulate **X**.
**T** = **R**_*True*_**C**_*True*_(8)
**X** = **T** + *Noise*(9)

The linear model assumed by Nous specifies that each dimension (column in **R***_True_*, row in **C***_True_*) be linearly independent of the others, that they be allowed to contain a mix of positive and negative numbers without zero values, that all other possible dimensions contain only zero, that all nonzero dimensions form a contiguous block of width (dimensionality or rank) *D*. **T** represents the “world as it truly is” prior to observation. To **T** is added a “noise” matrix of Gaussian random numbers representing observational error to obtain matrix **X**, the “observed” data matrix, from which cell values may be missing. It is understood that there may be different degrees of noise per row and column. **X** is the data we are presented with. We have no idea what **R***_True_*, **C***_True_*, and **T** are. After determining best dimensionality, a matrix decomposition algorithm like alternating least squares is applied to **X** to calculate coordinate arrays **R***_Est_*, **C***_Est_* (which will look nothing like **R***_True_* and **C***_True_*, except for their shape), and these will multiply to create estimates matrix **E**. We hope that **E** is as close as possible to **T**. Given this conceptual schema, there are three levels of error:

*Raw Residuals:* **Res** = **X** − **E**. The observed signed (positive or negative) discrepancies between observed and expected values, available only for non-missing cell values. It is the squares of these values that are minimized by alternating least squares.

*Expected Absolute Residuals:* **EAR.** Each **EAR***_ni_* can be thought of as the standard deviation of a hypothetical distribution of estimates for cell **X***_ni_*, an estimation of |**X***_ni_* − **E***_ni_*|. (Bootstrapping develops a literal distribution of cell estimates using repeated sampling and estimation.)

*Standard Errors:* **SE**. Each **SE***_ni_* can be thought of as estimating |**E***_ni_* − **T***_ni_*|, the expected absolute difference between a cell estimate and the corresponding unknown “true” value for that cell. This is the sense in which “standard error” is used in this paper. It is not the expected distance between the estimates and *observations* (the **EAR**) but between the estimates and *true values*. We are trying to answer the question for each cell estimate (regardless of whether its cell has a data value), “How *true* is this estimate?”

#### 2.6.1. Expected Absolute Residuals

Nous avoids the problem of repeated sampling by treating the residuals matrix, more particularly the squared residuals, as a decomposable matrix in its own right. The same alternating least squares (ALS) algorithm that was applied to data matrix **X** is now applied to squared residuals matrix **Res**^2^. We at first assume a dimensionality of 1 on the theory that each squared residual can be modeled as the product of a single row and column parameter.
**EAR**^2^ **=** *ALS*(**Res**^2^, ndim = 1)(10)

However, the actual decomposition is complicated by the fact that squared residuals have a hard floor of zero, indicating a ratio scale, whereas alternating least squares assumes interval data with no hard floor or ceiling. This is addressed by first taking the log of each squared residual, which converts it to an unbounded interval metric. However, taking the log of a product converts it to a sum, whereas our matrix decomposition assumes we are multiplying a row parameter by a column parameter. This is addressed by specifying a second dimension (because dimensions are additive in a dot product), so that each cell is treated as the dot product of an *N ×* 2 **R** coordinate array and a 2 *× I* **C** coordinate array. When **R** and **C** have been calculated with alternating least squares, their matrix product is an estimated squared residuals matrix in a log metric. This is converted back to the raw squared residuals metric by taking the exponent of the estimates and using polynomial best fit (degree = 1, which finds an optimal slope and intercept term), or “polyfit”, to maximize the column-by-column fit between the estimated and raw squared residuals. We take the square root of the resulting matrix to obtain the expected absolute residuals matrix.
**EAR** = *polyfit*(*exp*(*ALS*(*log*(**Res**^2^), ndim = 2)))^(1/2)^(11)

The predictive power of the **EAR** model relies on the idea that there is a characteristic degree of noise variance for each row and column, that a person who is, say, quick and careless for one set of items will be equally quick and careless for them all or that an item that is confusing for one set of persons will be equally confusing for them all. The assumption of characteristic noise per row and column can never be completely true in practice, and to that degree the cell expected absolute residual will be under- or over-estimated, but it will generally be true, and in principle, a secondary analysis of fit of expected versus observed residuals is possible that would flag persons and items with erratic variance, for whatever that is worth.

#### 2.6.2. Standard Errors

As mentioned, the expected absolute residual **EAR** measures the expected variance between estimates and observed values. It does not measure the expected variance between estimates and true values. This is easily observed in simulation studies in which the “true” values **T** can be compared to estimates **E**. However, in real life, **T** is never observed. How then estimate the “residuals” between **T** and **E**? It is at this point that we offer a formula for matrix **SE** given matrix **EAR**, based loosely on the classical formula for standard error, *SE* given the standard deviation *SD* of a single column of data with count *n*:(12)$$SE=\frac{SD}{\sqrt{n}}$$

In place of *SD*, we use **EAR***_ni_*, which can be interpreted as the expected standard deviation of estimates for cell *ni*. In place of *n*, we use a factor that is related to the number of observations *r* in row *n*, the number of observations *c* in column *i*, and the number of dimensions *d* in **R** and **C**:(13)SEni=2 ∗EARnird−112∗cd−11212

Alternatively:(14)SEni=2 ∗EARni∗d1/2r−d∗c−d14

A proof of this formula has not, so far as we know, been derived; our claim of its validity rests on simulation studies comparing the standard error prediction of the formula to the simulated |**T***_ni_* − **E***_ni_*|. However, the formula makes a certain amount of intuitive sense. We have already mentioned that **EAR** can be thought of as similar to the standard deviation of estimates (assuming repeated sampling) per cell, like *SD* in the classical standard error formula. Now, consider that when the number of columns (or rows, whichever is smaller) equals the number of dimensions, the estimates matrix **E** calculated using matrix decomposition will exactly equal the observations matrix **X**, no matter how much noise or disturbance **X** contains. Although **E** may be valid (depending on the validity of **X**), the uncertainty of **E** is essentially infinite because we do not know the validity of **X**, which it mimics. Therefore, standard error **SE***_ni_* should go to infinity if either *r* = *d* or *c* = *d*. The formula reflects this behavior in the structure of its denominator. As either condition *r* = *d* or *c* = *d* is approached, the denominator approaches zero, and the standard error approaches infinity. Meanwhile, the square root of *d* in the numerator of Equation (14) reflects how the number of dimensions, or degrees of freedom, directly magnifies the standard error. The multiplier 2 reflects, we believe, the number of “facets” of the data matrix, that it is a multiplication of *two* matrices **R** and **C**. A generalization of the model to handle more than two facets, for example a dataset whose values represent the interaction of a person, an item, and a rater, would presumably see this factor increase to 3 with corresponding changes in some of the exponents, but that has not been explored. When there is only one facet and one dimension, Equation (14) would reduce to Equation (12), the classic standard error formula, except that the denominator becomes *sqrt*(*n* − 1) rather than *sqrt*(*n*). This reflects the position that when there is only one data value in a sample, its standard error should be considered infinite.

The formula performs well in simulation studies across a wide range of variables and conditions. It even performs well with dichotomous and polytomous data. This may be surprising in light of the fact that dichotomous data do not meet the requirement for homoscedasticity of error, which is required by ordinary least squares. However, as mentioned, the solutions (**R**, **C**) of ordinary least squares, unlike maximum likelihood estimation, are not biased by heteroscedasticity of error. Because the estimation of the **EAR** and **ESE** matrices (see below) are based on such least squares solutions, they are also unbiased.

However, it is unlikely that Equation (14) is complete. For example, there is a persistent nonlinear relationship between true noise and estimated standard error, causing mild overestimation of error when noise is small and underestimation of error as noise becomes very large. Also, of great importance, it is only valid when the dimensionality is correctly specified.

#### 2.6.3. Expected Standard Errors

In principle, Equation (14) is sufficient to estimate cell standard errors, but Nous goes an extra step by treating matrix **SE** as decomposable and applying alternating least squares to compute its **R** and **C** coordinates and Expected Standard Errors **ESE** matrix using exactly the same procedure as for Equation (11), but applied to the **SE** matrix:**ESE** = *polyfit*(*exp*(*ALS*(*log*(**SE**^2^), ndim = 2)))^(1/2)^(15)

It may be wondered whether the validity of this procedure is compromised by missing data, in particular whether variation in the number of missing cells per row and column violates the model requirement of a “characteristic” degree of noise per row and column. It does not. The “missingness” of a given row will have a constant effect on each cell standard error in that row via the denominator in Equation (10), causing “missingness” to be folded into the standard error coordinates for that row. The same applies to the columns. Thus, the interpretation of standard error is always a combination of compliance with the model (**EAR**) and the count of independent data values relative to dimensions used to generate an estimate for that cell (*r*, c, and *d*).

The chief advantage of this extra step to calculate **ESE** is that it generates a set of **R***_SE_* and **C***_SE_* coordinates that concisely summarize matrix **SE**. Thus, in total, Nous generates three sets of **R** and **C** coordinates: (1) **RC***_Est_* that models observations to produce cell estimates; (2) **RC***_EAR_* that models residuals to get the expected distance between estimates and observations; and (3) **RC***_SE_* that models the calculation of cell standard errors to obtain the expected distance between cell estimates and unobserved “true” values. This makes it possible to attach the three sets of coordinates to each person and item in a way that is truly portable and that supports abstraction away from the original dataset, so that one can calculate person and item standard errors on the fly with no more than a single vector of data. Such portability is an important benefit of the objectivity property, applying not just to measurement and prediction but to the errors around those measurements and predictions.

### 2.7. Accounting for Age

If a young respondent receives a low score, is it because the respondent is young or because he or she has dyslexia? It is essential to work this out. The expert clinician who is familiar with the age of each respondent has, in principle, taken age into account in assigning a dyslexia rating. The successful prediction of clinician ratings under this assumption would indicate that the items and their decomposition are somehow able to disentangle age from dyslexia. While possible in theory, it is not to be taken for granted. It was decided that age should be separately modeled and explicitly distinguished from the dyslexia dimensions. This was achieved by performing a multistep factorization as follows:Log age. The natural log of each person’s reported age was calculated, which has the effect of more or less equalizing the effect of age differences across the age range, a requirement of least squares.First factorization. The *log*(age) variable was “anchored” as the first dimensional coordinate **R**[1]. Applying least squares, each set of column coordinates **C**[*i*] for each column *i* was calculated using **R**[1] and the observed (but normalized) data **X**[*i*]. This resulted in the first dimension of column coordinates **C**[1] and cell estimates matrix **E**[1] (the matrix product of **R**[1] and **C**[1]), as well as residual matrix **Res** = **X** − **E**[1].Second factorization. Nous was then applied to residual matrix **Res** to obtain the optimal dimensionality *D* and coordinates **R**[2→*D*] and **C**[2→*D*]. *D* is the number of dyslexia and other dimensions, the age dimension having been stripped out by the first factorization.Merging **R**[1] and **R**[2→*D*]. The dyslexia dimensions were appended to the age dimension **R**[1] to create an (almost) final version of **R**; call it **R**[*penultimate*].Finalizing **C** and **R**. Anchoring on **R**[*penultimate*], least squares was used to calculate a final version of **C**, and **C** to calculate the final version of **R**. This caused the person coordinates to be calculated last in the iterative process, ensuring that the person measures reflected the best possible fit between **X** and **E**.

Thus, accounting for age was achieved by explicitly assigning age to one of the **R** dimensions and allowing Nous to calculate the remaining dyslexia dimensions by applying the regular factorization procedure to the residuals matrix **Res**. The effect of this procedure was to provide age-adjusted “bumps” to the success rates of younger examinees (resulting in lower dyslexia risk scores), bumps already presumably applied as part of the clinical evaluation, enabling Nous to model the clinician ratings more precisely.

### 2.8. Setting Cut Points

The decision regarding where on the DRS to set cut points demarcating a *none*, *low*, *moderate*, and *severe* risk of dyslexia is a matter of clinical judgement. In our case, the clinician assigned three such cut-points based primarily on a personal examination of individuals in the calibration sample and secondarily, on the shape of the score distribution. It is possible that other clinicians would set the cut points differently. Ultimately, a panel of experts should make the decision based on community norms.

### 2.9. Scoring Individuals

Although the factorization procedure described above yields dyslexia risk measures for each individual in the dataset, the practical problem arises of how to calculate a measure for each additional person who takes the instrument. It would be unwieldy to redo the entire analysis every time a new person is added, especially because scoring must be performed in real time online or on a computing device. Coordinate anchoring offers a convenient solution. The analysis above, including analysis of fit, is characterized as “calibrating” the items in the dataset, deriving coordinates for all items as well as the clinician rating, and saving them in an “item bank” for future reference. Similar coordinates are saved to calculate person **EAR** and **SE** statistics on the fly. This is the “calibration phase”.

The “scoring phase” occurs when new person *n* takes an instrument that includes some subset of items (*I* > *D*) in the bank. The items to which person *n* responds are matched against those in the item bank, and the corresponding coordinates captured **C**[*matched*]. These are used to predict person *n*’s responses (normalized to pseudo-logits) to calculate **R***_n_*—person *n*’s coordinates in the common space. The dot product of **R***_n_* and clinician coordinates **C**[*cl*] from the calibration phase yields a preliminary dyslexia risk measure that is rescaled to fit on the 0–100 DRS scale and assigned a dyslexia risk category.

Each DRS score is accompanied by a dyslexia risk classification (*none*, *low*, *moderate*, *severe*), a person-specific standard error statistic, a fit statistic, and a flag indicating whether the score is “interpretable”. The interpretability of a person score depends on whether the person’s root mean square misfit statistic across responses is less than 2.0, where the cell misfit is:**Misfit***_ni_* = (**X***_ni_* − **E***_ni_*)/**EAR***_ni_*(16)

The possibility of deciding the interpretability of a person score is not often discussed in the psychometric literature, but it is implicit in Rasch modeling. When a person’s responses show substantial misfit to the model, it indicates the person has an indeterminate or contradictory location in the psychometric space; the score is, to that degree, meaningless. The proper procedure is to flag the examinee for personal attention and set their score aside.

The procedure for calculating a DRS score and standard error and misfit statistics is efficient, requiring only one person’s responses, identifiers of the items taken, and the pre-calibrated item bank of coordinates. Examinee age is not required for individual scoring because the estimated age dimension across items is sufficient for the algorithm to infer the age of the examinee implicitly and compensate accordingly.

## 3. Results

### 3.1. Optimal Dimensionality

Dimensionality analysis was performed to calculate accuracy, stability, and objectivity statistics for each of a sample of 10 dimensions (Figure 3). It found an optimal dimensionality of 5, not counting the precalculated age dimension—6 dimensions in total.

Examination of the accuracy curve (Equation (5)) shows that the ability to predict pseudo-missing cells rises with dimensionality and hits a plateau at dimension 5 (accuracy = 0.51). Examination of the stability curve (Equation (6)) shows that the cross-correlation of **R** calculated using two-item samples starts off high, declines slowly until dimension 4, then drops off slightly more quickly. The objectivity curve, the root product of accuracy and stability (Equation (7)), rises like the accuracy curve, peaks at dimension 5 (objectivity = 0.67), and declines slowly like the stability curve. Thus, the optimal dimensionality for this dataset is 5 + 1 = 6, the five dyslexia dimensions plus the precalculated age dimension.

The stability curve for a perfectly within-item multidimensional dataset usually starts as a slightly rising plateau up to the optimal dimensionality, then declines quickly. The fact that ours declines starting with dimension 1 suggests that the dataset is not strictly within-item multidimensional; items are not all sensitive to exactly the same 6 dimensions. A separate analysis of the non-timing items suggests that the timing items contribute extra dimensions to the dyslexia space, likely the main violator of the within-item multidimensionality specification. In effect, the timing items make it possible to partition the sample into four kinds of respondents: (a) slow and low-scoring; (b) slow and high-scoring; (c) fast and low-scoring; (d) fast and high-scoring. Such distinctions help disentangle dyslexic from non-dyslexic response patterns and contribute to accurate risk assessment but at the expense of some within-item dimensional consistency.

This raises the question of whether the two types of items should be analyzed separately then combined or the timing items should be dropped entirely. We decided that the timing items add essential information about dyslexia risk and that there is sufficient overlap in the dimensionality of the two item types to justify combining them in one analysis. The cross-correlation stability statistic at dimension 5 (stability = 0.89) is, in any case, sufficient for a reasonable expectation of reproducibility across item subsets, particularly if they include a stable mix of response items and their timings. However, the question may need to be revisited.

This paper spends little time trying to *interpret* the dimensions, an activity that is central to such techniques as factor analysis. There are several reasons for this. First, the activity is difficult and often inconclusive. In this case, we know that one of the six dimensions is age. From experience, we know that one of the dimensions is a technical adjustment relating to the “difficulty” of each item, and that an additional technical dimension or two is needed to model the dichotomous data. There is almost certainly a dimension reflecting the difference between the response items and their timings. That leaves two or three dimensions to model the dyslexic space itself. With some effort, guided by the extensive research in the field, it may be possible to assign names to those dimensions. The identification of dyslexia dimensions is certainly helpful for designing new items and securing an expectation that all participate in a common space. But from a measurement and prediction point of view, identifying and interpreting dimensions serves little purpose. While the number of dimensions making up a space is of utmost importance, the orientation of the axes of that space and the placement of the origin point are entirely arbitrary. The coordinates are not human-interpretable and probably should not be. What does need to be human-interpretable are the items and clinician ratings. These are defined by humans for humans and reflect a human judgment of what dyslexia is and how it should be measured.

### 3.2. Analysis of Fit

Prior to performing analysis of fit, the “outfit” statistic averaged across all items was 1.24, where 1.0 is ideal and 2.0 indicates “statistically significant misfit overall” (at the 95% confidence level). Cells with misfit > 2.0 comprised 12% of the dataset, in which 5% would be expected by chance. Removing from analysis various items and persons that did not fit the 6-dimensional space, it was found possible to improve reliability, objectivity, and fit. Well-fitting and reliable solutions were found by removing as many as 100 items and persons.

At the same time, we were particularly interested in the fit of the clinical ratings to the model predictions because these are the basis of the DRS. In this case, the column misfit statistic was 0.996, with 8% of cells with misfit > 2.0, signifying almost perfect fit relative to the expected absolute residual for that column.

For this and other reasons, it was decided that there was no compelling reason to remove misfitting items on theoretical grounds (except for one category of timing items that was found to contain a data collection error), that improvements from dropping misfitting items and persons were minor, and that no important loss would be incurred by using (almost) the full dataset to calibrate the item bank.

The number of persons with sufficient misfit to warrant a classification of “uninterpretable” ranged from 1 to 10% depending on how low the misfit threshold was set.

### 3.3. Age and Item Drift

Because of the importance of removing age as a disturbing factor in dyslexia risk measurement, tests were performed to confirm that the age factor was correctly modeled. One test was to break the dataset into three groups by age—“children” (aged 7–9), “teens” (aged 10–15), “adults” (aged 16 and above)—and analyze each group separately. This was achieved by calibrating the items separately for each group and comparing the resulting item coordinates for “drift”—similar to what is called “DIF” (differential item functioning) in educational psychometrics. Item drift is the degree to which the item coordinates fail to correlate across person groups. We found that while there might be minor statistical benefits to analyzing age groups separately, the item drift statistics were low and the coordinate reproducibility was high (*r* > 0.97) across age groups, which was sufficient to justify analyzing all ages together in one common coordinate system. This supports the idea of a single dyslexia risk scale applicable for all ages. 

### 3.4. Dyslexia Risk Scale Distribution

The distribution of dyslexia risk scores for the sample of 814 persons has, as expected, an elongated upper tail reflecting the selection of moderate to severe dyslexic individuals in the calibrating sample (Figure 4). Table 1 provides distribution descriptive statistics.

The assignment of cut scores was performed by the clinician by identifying examinees who, in his judgment, lay on the boundary between no and low dyslexia risk, low and moderate risk, and moderate and severe risk, reviewing their DRS scores, and specifying cut scores that were close to those scores. Table 2 was the result.

The center of the DRS (50) corresponds approximately to the median of the distribution and separates those who have low or no risk of dyslexia from those with moderate to severe dyslexia. The moderate-to-severe cut score (55) corresponds approximately to the 75th percentile, and the none-to-low cut score (45) corresponds approximately to the 25th percentile. While it is convenient that the cut scores happen to correspond with the statistical properties of the distribution, they were not chosen for that reason; they primarily reflect a symptom-based clinical judgment on the basis of which intervention decisions can be made. While other clinicians reviewing the same data might assign cut scores differently, these seem defensible as a first step to a more formal collective standard setting.

### 3.5. Do DRS Scores Predict Clinician Ratings?

One way to think of the DRS is as a tool for predicting, in a clinician’s absence, how the clinician *would* rate an examinee on a 0–100 linear scale. Because the procedure described in Section 2.9 implements an efficient scoring process, it becomes possible for individuals who need an accurate risk assessment to obtain one at minimal time and expense. But does the DRS work as a reasonable proxy for clinician judgment? To answer this question, we first plot the DRS against clinician ratings (Figure 5):

Aside from several outliers in which the DRS measure is markedly lower than the clinician rating, the pattern is what one would expect: vertical bands for each clinician rating due to the discreteness of the rating scale, a positive but nonlinear relationship, a plateau in the middle and separation at the extremes hinting at a vertical ogival relationship. The Spearman correlation is *r* = 0.54. 

To quantify the predictive power of the DRS more precisely, we performed a logistic regression with the DRS as the sole predictor variable (plus a constant) of a [0,1] binary clinician categorization of severe dyslexic risk, in which “1” was defined to be a rating of 7 or above (the clinician’s flag for actionable dyslexia risk). To strip away any bias toward the clinician ratings (overfit), the DRS was based on an **R** coordinates matrix calculated without the clinician ratings column (the bias was minimal in any case). The DRS predictor coefficient was 0.6810 with 95% CI [0.57, 0.79], *SE* = 0.055, *z* = 12.372, *p* < 0.001, indicating a strong and statistically significant predictive relationship. The pseudo-R^2^ (McFadden’s variance explained) was 0.6524. Though mainly used for relative comparisons, McFadden offers as a rule of thumb that the pseudo-R^2^ tends to run lower than the ordinary least squares R^2^, and values between 0.2 and 0.4 represent excellent fit [23]. 

The predictive accuracy of the model using the DRS 50 cut point (“moderate or severe”) is 91% with a sensitivity of 91% (proportion of true positives to all actual positives) and a specificity of 91% (proportion of true negatives to all actual negatives). Because the DRS 50 cut point generates a favorable balance between sensitivity and specificity while at the same time generating a sensitivity that is sufficient to flag examinees with moderate to severe dyslexia risk, we saw no need to adjust the DRS 50 cut point to increase sensitivity. Note that a DRS cut point of 50 achieves this balance of sensitivity and specificity only when predicting a clinician rating of 7 or above; the DRS and the clinician rating scale are not linearly related. Thus, a DRS finding of “moderate or severe” (>50) is ideally predictive of a clinician rating of 7 or above.

To examine the tradeoff between sensitivity and specificity more closely and evaluate the discriminative power of the DRS across the range of possible cut scores, we plotted the “true positive rate” (*sensitivity*, *y*-axis) against the “false positive rate” (1—*specificity*, *x*-axis) to obtain a receiver operating characteristic (ROC) curve (Figure 6).

If the proportion of the area under the curve (AUC) is 1.0, all positive diagnoses are correct with zero false positives—the test is maximally discriminating. An AUC of 0.50 indicates no discrimination. While there is no single accepted standard for minimum AUC, a finding of 0.90 or above is considered excellent according to the University of Nebraska Medical Center’s guidelines for evaluating the accuracy of diagnostic tests [24]. The DRS yields an AUC of 0.96 when the clinician rating cut score is 7. For comparison, Nergård-Nilssen and Eklund [25] report an AUC of 0.92 for the 6-part “Norwegian screening test for dyslexia”. The symmetry of the ROC curve about the principal diagonal illustrates a favorable balance between sensitivity and specificity when the sensitivity is near 0.90.

## 4. Discussion

Identifying individuals at risk for dyslexia-related challenges at school or work is essential to helping these individuals reach their full potential. In recent years, emphasis has justifiably been placed on the early identification of at-risk children, with the greatest focus on ages 4–7. While these efforts are invaluable, the needs of older at-risk individuals, not only of school age but throughout adulthood, must also be addressed.

The dyslexia screening tool described here enables the identification of at-risk individuals aged 7 and above, with a reliability of 0.95, a predictive accuracy of 91%, and an area under the ROC curve of 0.96. Furthermore, the analytic method presented here, by adapting matrix factorization to Rasch-like criteria for measurement objectivity, offers a clear and reproducible procedure for placing the measurement of dyslexia risk on a stable conceptual and measurement foundation on par with large-scale state education assessments. However, given the novelty of this approach for the domain of cognitive disorders, the truth value of these measurements needs to be considered beyond the simple statistical question of how well they align with one clinician’s ratings. There are two questions in particular that need to be faced: (1) Are DRS scores possibly more true than the original clinician ratings? (2) Are DRS scores likely to be more true than those of existing instruments?

### 4.1. Are DRS Scores “More True” Than Clinician Ratings?

If we assume the clinician assessment to be correct in all cases, the DRS (calculated without the assistance of a human clinician) can, based on the logistic regression above, be considered “correct” approximately 91% of the time and may justifiably be claimed as sufficient to stand in for the clinician. However, clinical diagnoses are not always correct due to human fallibility and the difficulty of defining and maintaining a consistent standard, especially in a multidimensional space. Are there grounds for claiming the DRS scores to be *more true* on average than the clinician ratings on which they are based?

In the absence of a standard error statistic for clinician ratings, no direct comparison with the DRS root mean square error (RMSE) is available. However, Figure 4 shows that DRS variation *within* a rating scale category tends to be at least 10 DRS points, often much more. With an RMSE of 2.38 (Table 3), that means variation within a rating scale category will tend to range across three standard errors, indicating that three strata of examinees within a scoring category are statistically distinguishable from each other (68% CI) and that, therefore, the DRS in this limited sense is more precise and fine-grained than raw clinician ratings.

Information theory offers a clearer way to compare the clinician ratings with their corresponding DRS scores, in terms of their information entropy:(17)$$Entropy=-\sum_{i}^{I}p_{i}{log}_{2}p_{i}$$

Assuming each possible clinician rating has a probability *p_i_* corresponding to its frequency in the study sample distribution, the clinician rating Shannon entropy is 3.27. Because DRS scores are continuous, we partition the scale into equal-length standard error units of 2.19 DRS points, the median standard error of the sample. This results in 30 DRS bins or scoring categories versus 11 clinician ratings. From the proportion of persons in each bin, probability *p_i_* is calculated, and the DRS Shannon entropy comes to 3.94, which exceeds the clinician rating entropy by 0.67.

Because the person distribution in this study is not representative of the general population and was, in fact, designed to have a roughly equal proportion of dyslexic and non-dyslexic examinees, it may make more sense to calculate entropy assuming an equiprobable distribution, in which no single dyslexia rating or measure is more or less probable than any other. In this construction, the clinician entropy is 3.46, and the DRS entropy is 4.81, 1.35 points higher, reflecting the fact that the number of information bins has more than doubled (each additional point on the Shannon entropy scale represents a doubling).

These calculations assume that a difference of one standard error is sufficient to differentiate two adjacent scores, a purely human convention. Perhaps a bin size of two standard errors is needed, or perhaps only half is needed. The smaller the designated standard error unit, the larger the DRS entropy. In any event, with a standard error unit of 2.19 DRS points or smaller, the DRS information entropy exceeds that of the clinician ratings.

That should come as no surprise. DRS scores pull information in a consistent way from all 400 or so examinee responses and timings, stripping away the multidimensional peculiarities of each item. The result, taking into account statistical noise and the 6-fold dimensionality of the instrument, is a separation statistic of 4.53 and reliability of 0.95, on par with (or better than) the reliability statistics of many large-scale educational assessments (Table 3). “Separation” is a function of the ratio of the standard deviation of the person sample to the RMSE (measurement error per person); it plays a role in psychometrics that is conceptually analogous to entropy in information theory. “Reliability” converts separation to a statistic between zero and one and is very similar to the Cronbach-alpha statistic. That means DRS scores offer an improvement on clinician ratings that is comparable to what one might expect of large-scale assessment scores over teacher ratings of the same students.

Because reliability is a function of number of items, it may be wondered why the DRS reliability is not, perhaps, even higher given the length of the instrument. The answer lies in the dimensionality, a key element in the standard error formula (Equation (14)). If each dimension is thought of as its own orthogonal test dimension, then there are roughly 74 items per dimension, which is more in line with what one would expect given the 0.95 reliability statistic.

The “accuracy” statistic (Table 3), as discussed above, refers to the (Pearson) correlation between observed values for pseudo-missing cells throughout the dataset and the corresponding cell estimates (*r* = 0.51). The model does a reasonable job of predicting missing cells across the dataset, and a better job of predicting dichotomized clinician ratings (*r* = 0.91). “Stability” is the Damon equivalent of the parallel forms cross-validation correlation statistic, also called inter-test reliability—how correlated the person coordinates calculated from two independent halves of the dataset are. This is fairly strong (*r* = 0.89). The “objectivity” statistic of 0.67 combines the two. 

While it seems clear that DRS measures are likely more precise and reliable than clinician ratings and contain more information, that does not mean they are “more true”. That depends on the degree to which clinician ratings include insights or procedures that cannot be captured using a linear model or that require dimensions beyond the six that have been identified. For instance, if for illustrative purposes, we suppose that a clinician employs as a mental model for high dyslexia a person with “50th percentile dyslexia” and assigns ratings based on *proximity* to that ideal (which would not make sense in practice), a linear model would fail because a person in the 75th percentile would have a similar rating as someone in the 25th percentile due to the nonlinearity of “proximity”. 

However, if the model person is conceived to have “95th percentile dyslexia” (a more likely scenario), a linear model would be largely successful because proximity to that person would, in 95% of cases, imply having more dyslexia. As long as a clinician’s rating can be visualized as projecting persons onto a straight-line yardstick with increasing increments that spans the entire 6-dimensional person distribution, and no other dimensions, those ratings should be predictable within the proposed linear matrix factorization model. That this is true of the current clinician ratings can be inferred from the DRS/clinician rating misfit statistic of 0.996. Because the ratio between observed and expected variance for cells in the clinician rating column is almost 1.0, we can consider the two as essentially equivalent; the cell residuals are no larger than what one would expect by chance. At the same time, the reliability statistic (0.95) is within an acceptable range to support practical measurement, indicating that noise is not excessive. Therefore, the DRS measures can be viewed as “true” (but more precise and reliable) to the degree the clinician ratings are “true”.

### 4.2. Are DRS Scores “More True” Than Those of Existing Instruments?

This question must fall under the category of “future research”. No comparisons with existing dyslexia instruments, such as dyslexia screeners currently used by school districts to flag students, have yet been attempted. However, there are grounds for suspecting that when such comparisons are performed, the DRS scores will *not* be found to be highly correlated to existing instruments. Most instruments have less than 50 items, with no timing items, and are analyzed implicitly assuming unidimensionality. The DRS has roughly 443 items per person, includes timings, and locates persons in an empirically determined 6-dimensional space. This is virtually an order of magnitude more data per person, and those data take into account how quickly a student responds to an item and the dimensionality of his or her responses. Given the complexity of dyslexia, it would not be surprising to find that this more microscopic approach to measurement would yield fairly different scores. For example, the correlation between DRS scores and the initial clinician/self-report ratings used to select respondents is only 0.54, implying that the DRS finds indications of dyslexia that are hard to discern based on surface-level self-report or clinical inspection.

In the absence of a gold standard for evaluating dyslexia, there is no straightforward way to claim, aside from statistical measures such as reliability, that one instrument yields scores that are “more true” than another. One must rely on the degree to which an instrument embodies the dyslexia construct as it is understood by the dyslexia research and practitioner community. Having a panel of dyslexia experts evaluate the items in the DRS for relevance to dyslexia and assign its own ratings to a representative sample of persons, while assigning clinically useful cut points, would go a long way toward constructing such a gold standard and evaluating the degree to which the procedure presented here, and other instruments and procedures, yield measures that reflect that standard.

### 4.3. Where Does the Extra Information Come from?

Looking up from the weeds of dyslexia diagnosis, it may be of interest to view matrix factorization from a high-level information-theoretic perspective. Consider a raw data matrix **X** consisting of dichotomous values that presents itself for analysis. We run it through a black box analysis algorithm like Nous and obtain a new matrix **E** that is exactly the same size as **X** but with this difference: the information entropy of **E**, calculated with the help of a standard error statistic as discussed in Section 4.1, is considerably higher than that of **X**. No new information was added by Nous, yet its output has higher information content or capacity. How is this possible? Where did the extra information come from?

This is not a trivial question. We can say that the information was always there in **X**; we just stripped away the noise. And that is true, but *how* was the noise stripped away from something like a dichotomous [0,1] data value? Where did the additional information distinguishing noise from signal come from? We propose four sources:

*The Data Collector*. Had the data in **X** been arranged in a random fashion, tossed in the air and reassembled, it would not have been possible to calculate **E**. It is precisely the fact that each row is guaranteed by the data collector to correspond empirically to a single, specific examinee, and each column to a specific question, that makes the matrix decomposable. The data collector also sees that data are collected under standardized conditions so that each datum is comparable to every other. It is the assumption of a common “object” (a person or item) across a vector of data values that makes it possible to set up and solve the system of simultaneous equations that is the mathematical basis of Gaussian least squares. The arrangement of data into rows and columns is a human endeavor, managed through the data collection design, which transforms the human perception of commonality or identity across a multiplicity of data values into information.

*The Analyst*. The analyst does not treat all decompositions of **X** as having equal worth. Different dimensionalities are attempted; all but one is thrown out. That dimensionality is chosen that leads to estimates that best predict missing cells and coordinates that are maximally stable across samples. These criteria are not mathematically derivable; they reflect a human value called “objectivity”. Put differently, mathematics is used to find optimal objectivity but says nothing about whether objectivity is a desirable outcome. In addition, with Nous, the analyst has the labor of identifying and enforcing a common space, removing misfitting persons and items. Such activities add tangibly to the information content of **E**.

*The Item Writer*. Items are only mathematically tractable to the degree they behave the same way for each examinee. This requires considerable craft. In addition, items need to be written in such a way that they erect and participate in a common space of multiple dimensions. To the degree they fail to accomplish this, the model is compromised and error and misfit increase. 

*The Examinee*. The examinee has the hardest task of all. Each question requires an exertion of force that must be renewed not just once or a few times, but hundreds. Were examinees not willing to put in this effort, again and again, their responses would be mathematically intractable, showing up as person misfit and high error. It is the consistency of effort that makes it possible to treat their responses as the dependent variable of a system of simultaneous equations.

The extra information content of **E** over **X** is, in large part, a measure of the degree to which such human efforts and insights go into producing, collecting, and analyzing data. Note also that all these properties focus on a central mathematical property. The information available about an object is directly related to the degree it occupies one, and only one, position in space. Standard error, the prime contributor to lack of information, is a measure of the degree to which an object presents as dispersed over space, the degree to which a person, for example, appears to reside in different locations in space depending on the sample of items used to calculate his or her coordinates.

Geiger and Kubin [20] were cited earlier for their proofs showing that principal components analysis, and by extension, matrix factorization, minimizes the loss of information as dimensionality is reduced. The intuition behind such proofs should be evident from this paper. So long as the noise added to the hypothesized “true” matrix **T** (see Section 2.6) is truly random, and so long as the dimensions of **T** are truly independent without constraints on their values, **E** calculated by decomposing **X** at the correct dimensionality will be as close as possible not only to **X** but, through inference, to **T**, as well. The accuracy of **E**, its proximity to **T**, is then only a function of the amount of row and column data that is available, the number of dimensions, and the amount of noise added. This is concisely expressed in Equation (14), the standard error formula. 

It is important to realize that many datasets do not meet the mathematical requirements posed for **T**. For example, a matrix of distances is not strictly decomposable using matrix factorization because the Euclidean distance formula involves the addition of terms (dimensions) that are mathematically entangled. 

However, when the data properly reflect the **X** = **RC** + *noise* model, useful properties emerge, as has been pointed out. Persons and items can be treated as objects, their coordinates transferable to any other test that is sensitive to the same space, once their coordinate systems are equated. Under conditions of perfect fit, the three sets of matrices **R** and **C** (for the cell estimates, expected absolute residuals, and standard errors) summarize virtually everything that is worth knowing about **X**, i.e., that contains useful information. All else is, literally, noise. Thus, **R** and **C** can be viewed as a maximal form of data compression of **X**.

## 5. Conclusions

When receiving results from a state educational assessment, students, parents, and schools expect to receive, for each student, a reliable test score with a performance level indicating whether the student meets the state’s definition of “proficient”. These are expected to have a uniform meaning across schools and over time, and “proficient” is expected to have the same meaning for everybody. We believe that the analysis presented here, with its foundation in objective multidimensional measurement, offers an analogous workable prototype that, suitably generalized, would be sufficient to serve as a community gold standard for measuring dyslexia and other cognitive disorders.

Generalizing the procedure given here, one can imagine a dyslexia research collective of some sort that convenes a panel of experts to define what dyslexia is and works out specifications for items that are sensitive to each of a set of interrelated dyslexia constructs. This research collective would solicit, from practitioners, dyslexia items that meet those specifications and administer them to a wide and diverse sample of dyslexic and non-dyslexic examinees of all ages. A panel of expert clinicians would be convened to review the data and assign a “dyslexia risk” rating to each examinee. This would form the basis of a new scale. 

Matrix factorization would be performed to verify the dimensionality of the dataset, and the compliance of each item with that dimensionality, and to generate a continuous equal-interval dyslexia risk scale (plus any desired diagnostic subscales) that would have the same content validity as the ratings from the panel of expert clinicians but be more precise and reliable. The panel would then assign dyslexia risk cut scores to the scale. The calculated item and clinician rating coordinates (constituting an “item bank”) would be stored in an online database for live administration and scoring. 

Future examinees would take the test online or on a computing device and receive an immediate dyslexia risk score with a classification of “positive”, “negative”, or “uninterpretable”, with standard error, plus diagnostic subscale measures. The examinee’s clinician would then decide what interventions might be necessary with confidence that the diagnosis meets industry standards. This is now being performed by a private company, but it could be a service offered by the wider dyslexia research community.

For clinicians who have developed their own instruments, it would be fairly easy to equate their instruments to the collective item bank by administering their own instrument to a sample of persons along with items from the bank. An online calibration program would assign to the clinician’s proprietary items coordinates that reside in the same dyslexia coordinate system space as the collective items. From that point on, the clinician could, in principle, administer his or her own instrument yet obtain scores that are comparable, depending on item quantity and quality, to what *would have been assigned by the collective*. 

It would be straightforward to “grow” the item bank by integrating new items and data collected by practitioners in the field while retiring old items, making sure all are carefully vetted and fit within the specified dimensional space. Just such a collective approach has been employed since the 1980s in the educational field by the Northwest Evaluation Association (NWEA) [26]. By pooling items written by member school districts and equating them across tests and grades using a Rasch-based common-item design, NWEA has compiled a very large, regularly updated item bank to support administering computer adaptive tests online with immediate scoring on its RIT (“Rasch unit”) scale, in which each score is comparable to every other across grades and member districts, even when students take different items [27]. That has enabled member school districts to track educational outcomes and perform program evaluation studies to a degree many of their peers cannot. NWEA is effectively doing for educational measurement what laboratories and medical boards do for medical testing.

By extending this basic Rasch equating paradigm to multidimensional spaces using matrix factorization according to the principles outlined here, it is believed that participating dyslexia clinicians and their institutions can achieve similar results.

## Figures and Tables

**Figure 1 entropy-25-01580-f001:**
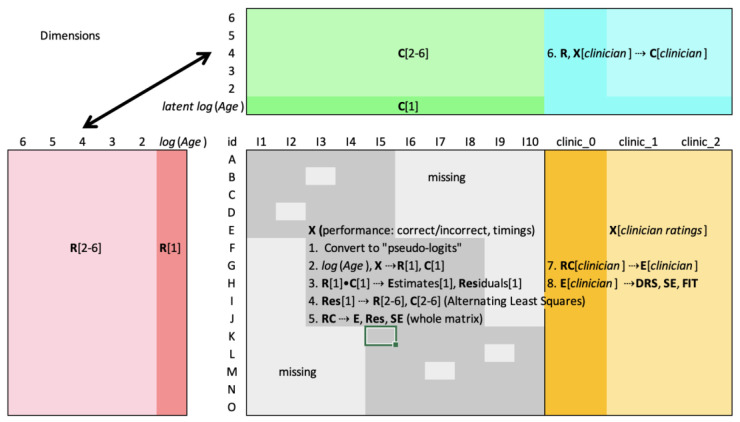
Matrix factorization and scoring procedure. Matrices **R** and **C** are calculated from **X** (the gray shaded cells), which includes blocks of missing data. Steps 1–5, accounting for age, are discussed in Section 2.7. Their product yields matrix **E**, whose values cover not only the cells with observed values (**X**) but the missing cells, as well. **R** also provides orthogonal predictor variables for calculating the clinician expected values that form the basis of the DRS by predicting the clinician ratings in the “clinic_0” column, as well as subscale ratings in the “clinic_1” and “clinic_2” columns, with standard errors and fit statistics.

**Figure 2 entropy-25-01580-f002:**
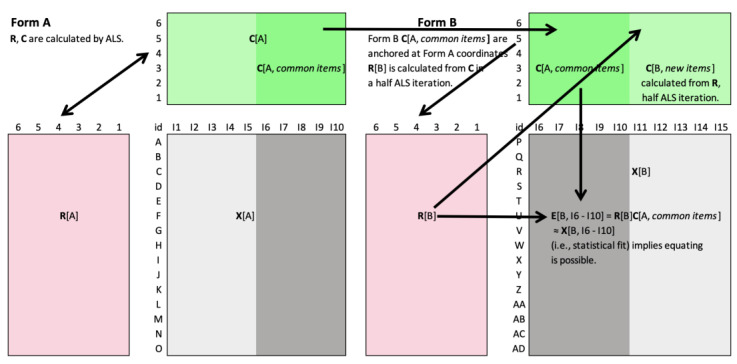
Equating Form B to Form A. Coordinates matrices **R**[A] and **C**[A] for Form A are calculated using alternating least squares. To equate Form B to A, the Form B **C**[A, *common items*] for the items the two forms have in common are “anchored” to their Form A values. The Form B **R**[B] coordinates are then calculated in a half iteration using **X**[B] data and **C**[A, *common items*]. The remaining item coordinates on Form B, **C**[B, *new items*], are calculated from **R**[B] and **X**[B]. If data **X**[B, I6-I10] for common items “fits” within statistical tolerances to the Form B estimates **E**[B, I6-I10] for those items (calculated using **C** coordinates from Form A), then we can claim the two forms are equated.

**Figure 3 entropy-25-01580-f003:**
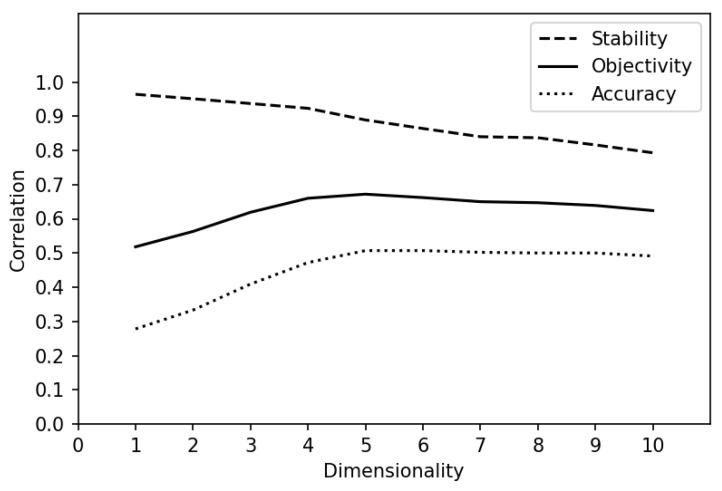
Objectivity curve. Objectivity, the root product of accuracy and stability, peaks at dimension 5, indicating five latent factors in the data, excluding the age factor.

**Figure 4 entropy-25-01580-f004:**
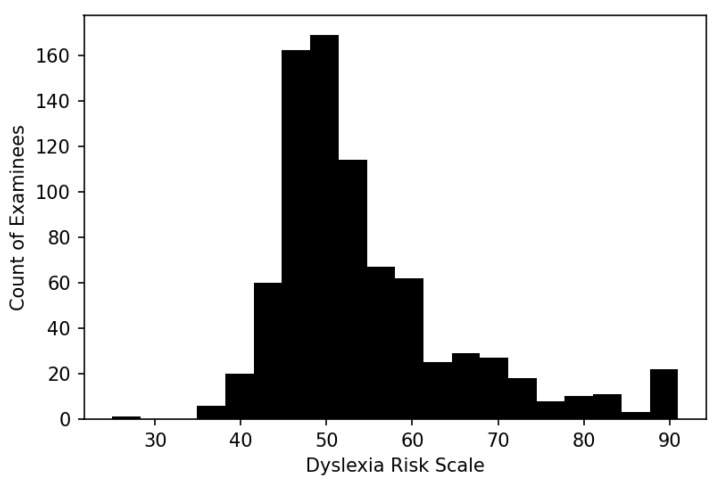
Dyslexia risk histogram (*n* = 814).

**Figure 5 entropy-25-01580-f005:**
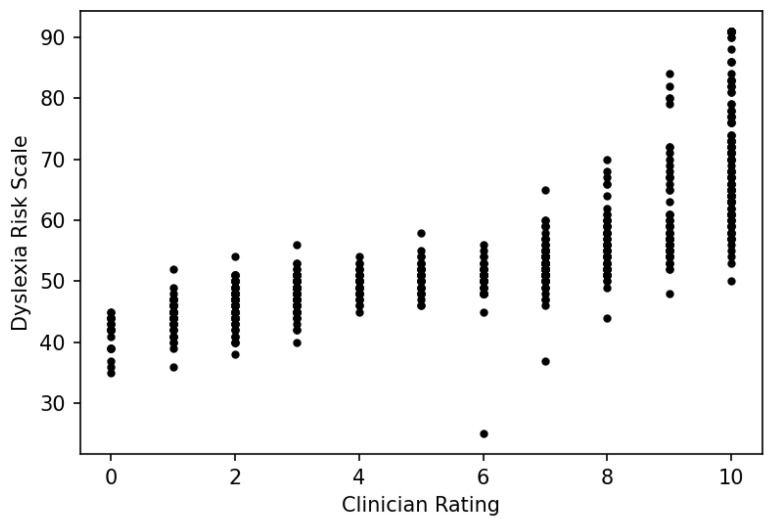
Dyslexia risk clinician rating vs. dyslexia risk measure (Spearman r = 0.54, *p* < 0.001).

**Figure 6 entropy-25-01580-f006:**
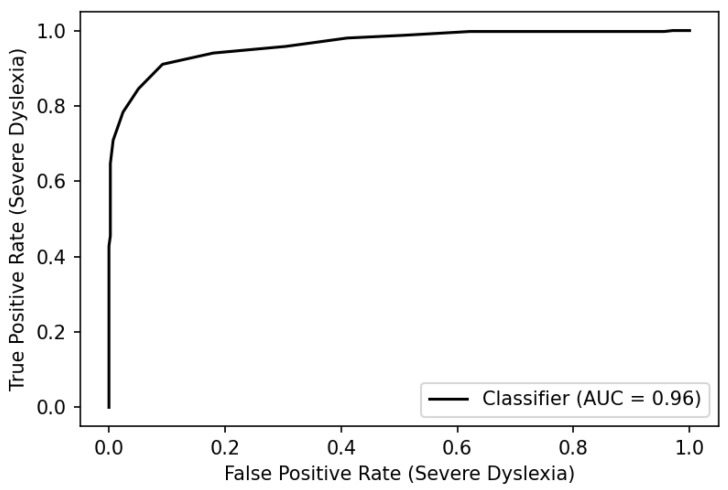
Receiver operating characteristic (ROC) curve, AUC = 0.96.

**Table 1 entropy-25-01580-t001:** Distribution statistics.

Statistic	Value
Count	814
Mean	54.5
Standard Deviation	11.0
Minimum	25
25th Percentile	48
Median	51
75th Percentile	57
Maximum	91

**Table 2 entropy-25-01580-t002:** Dyslexia risk scale cut scores.

Dyslexia Risk Category	Cut Score Range
None	0–45
Low	45–50
Moderate	50–55
Severe	55–100

**Table 3 entropy-25-01580-t003:** DRS statistical quality.

Statistic	Value
RMSE	2.38
Separation	4.53
Reliability	0.95
Mean Item Misfit (mean-square)	1.24
DRS/Clinician Rating Misfit (mean square)	1.00
Accuracy (cell value prediction)	0.51
Stability (parallel forms cross-validation)	0.89
Objectivity (Accuracy ∗ Stability)^(1/2)	0.67

## Data Availability

The code used to perform the analysis plus a *simulated* dataset are contained in a package called “dys_pack” located at https://sites.google.com/view/pythias-consulting/download (accessed on 31 October 2023). The actual dataset employed in the analysis consists of confidential patient data for which there is a legal expectation of confidentiality and cannot be posted publicly.

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
