# Peer review of "Information from Noise: Measuring Dyslexia Risk Using Rasch-like Matrix Factorization with a Procedure for Equating Instruments"

_entropy, 2023, doi:10.3390/e25121580_

Round 1

Reviewer 1 Report

Comments and Suggestions for Authors

This is an outstandingly innovative study. I think it will be of considerable interest to readers of the journal. I read it with great interest. Its originality is sky high, applying matrix factorisation methods to dyslexia diagnosis for the first time. It is also very strong methodologically - consider the time taken to undertake the 828 interviews on which the analysis is based!

I am very impressed by the very rare combination of the methodological rigour of the work undertaken together with the high technical level of understanding and analysis shown. Use of the entropy concept is of course a plus for readers of this journal! It is worth noting that this methodology could be used with benefit for a whole range of psychometric tests, with the Wechsler scales (IQ) being a natural one to explore since there is great interest in interpretation of such tests.

Author Response

No corrections suggested. The review is gratefully received.

Reviewer 2 Report

Comments and Suggestions for Authors

This study both examines the psychometric properties of a screening protocol for dyslexia and demonstrates a special form of matrix factorization called Nous based on the Alternating Least Squares algorithm. Overall, the contents of the manuscript are substantial, but it lacks logic and organization in its narrative, and its motivations and contributions are not prominent. In addition, the manuscript should be further improved in terms of writing. Here are some specific comments that may be helpful.

1. The Introduction section is too much space and it does not reflect the motivation as well as the innovation of the paper well. It is suggested that the authors can reorganize the Introduction section in the following ways:

(1) Enhance the logic and organization of the narrative;

(2) Some particularly basic knowledge, as well as irrelevant historical sources can be curtailed;

(3) For existing related work in the field, a separate section can be written if necessary;

(4) The motivation and contribution of this manuscript should be further highlighted.

2. Minor errors appear in the manuscript, such as "[they]" in the second paragraph of the Introduction section, and it is recommended that such errors be checked throughout the whole manuscript.

3. The apparent inconsistency in the representation of the matrix elements in Eq. (2) needs to be noted. Similar issues should be checked to ensure consistency in the representation before and after.

4. In Section 2.2, the proposed matrix factorization algorithm, i.e., Nous, is presented. However, the authors use a lot of textual content for the narrative, which is not easy to understand. It is recommended that the authors should provide a more detailed mathematical and rational presentation, while the rigor of the presentation needs to be ensured. It may be helpful to refer to the following literature on the mathematical and theoretical presentation:

FCAN-MOPSO: An Improved Fuzzy-based Graph Clustering Algorithm for Complex Networks with Multi-objective Particle Swarm Optimization.

5. In Section 2, a number of metrics for the assessment of uncertainty are presented and occupy a large amount of space. However, these metrics are not all presented in the Results section, so are these presentations redundant?

6. The layout of the manuscript regarding figures and tables should be further improved, especially the cross-page tables which are not recommended in general.

7. The Discussion section of the manuscript is not well-focused in its presentation, and it is recommended that the authors discuss and summarize the highlights of the experimental results and draw certain conclusions.

8. The Conclusion section of the paper should be concise and summarized, but the author's statement contains a lot of redundancy.

Author Response

We have attempted to respond to the points raised by this reviewer, making many substantial modifications which we believe improve the manuscript. We thank the reviewer for the attentive reading which has helped us make these improvements. The following are our responses to the specific reviewer points:

  1. We have attempted to clarify the logic and organization of the narrative, as well as its motivation and contribution, by adding a paragraph detailing explicitly the papers' objectives. We have also shortened the section by moving out a section more appropriately detailed in methods.
  2. The grammatical construction referred to in point 2 is not in error.
  3.  We have also corrected both textual and substantive errors, the most important of which required a recalculation of standard error and reliability (up from 0.94 to 0.95) after discovering a minor software error, and removing a paragraph about a binomial adjustment to error, which turned out to be unnecessary. 
  4. In section 2, while we were able to make some of the mathematical reasoning more explicit in the form of equations, much remains in textual form as a full mathematical justification and treatment seemed to require a level of elaboration that would add considerably to the length and complexity of the paper. However, sufficient detail is presented in the equations that another analyst should be able to reproduce our work.  
  5. We have not removed or shortened the "uncertainty" section for several reasons:  1) it is the mathematical basis of the separation, reliability, and entropy statistics presented in section 4; 2) it is highly germane to information theory; 3) this treatment of standard error is, so far as I know, original and might have real value for analysts and programmers working in the machine learning field, where error estimation tends to be problematic.
  6. We have attempted to improve the layout and appearance of the tables and figures.
  7. The Discussion section admittedly deviates somewhat from traditional academic paper templates, raising topics at a higher conceptual level than the detailed treatment of earlier sections.  This appears to us to be unavoidable.  This is not a specialty paper for a dyslexia or machine learning audience, nor even an information theory audience.  We have tried to indicate the possibility of a higher-order integration of these fields, which is where we feel the value of the paper lies. 
  8. The same explanation applies to the Conclusion, where we attempt to spell out what its implications might mean in practice for a field that, like many others in the health, psychology, and education world, has not progressed as much, psychometrically, as its importance warrants.

Reviewer 3 Report

Comments and Suggestions for Authors

I immensely liked this manuscript: it is a rare example of a scientific enterprise in which the authors, by means of a crystalline while rigorous style, drive the reader into the meaning of the proposed statistical techniques with a continuous reference to their clinical counterpart. The development of a sort of 'supervised PCA' able to deal with date sets plagued by missing data is a crucially important achievement that gives a direct solution to the 'explainability' issue that affects a great part of machine intelligence community. It is a solution that fosters a continuous exchange between experts in the field and data analysts and is a bright demonstration of how to develop a truly interdisciplinary science. It is not by chance that the same attitude was at the basis of the rise of modern statistics in the forst decades of the XX century.

Author Response

No corrections suggested. We thank the reviewer for the kind comments.

Round 2

Reviewer 2 Report

Comments and Suggestions for Authors

All of my concerns have been addressed.

Comments on the Quality of English Language

Good